# ZNF423 patient variants, truncations, and in-frame deletions in mice define an allele-dependent range of midline brain abnormalities

Ojas Deshpande[1,2], Raquel Z. Lara[1,2], Oliver R. Zhang[1,2¤], Dorothy Concepcion[1,2], Bruce A. Hamilton[1,2]*

1 Department of Cellular and Molecular Medicine, Institute for Genomic Medicine, Rebecca and John Moores UCSD Cancer Center, University of California, San Diego School of Medicine, La Jolla, CA, United States of America, 2 Department of Medicine, Institute for Genomic Medicine, Rebecca and John Moores UCSD Cancer Center, University of California, San Diego School of Medicine, Gilman Drive, La Jolla, CA, United States of America

¤ Current address: Rocky Vista University College of Osteopathic Medicine, Parker, CO, United States of America
* bah@health.ucsd.edu

**Data Availability Statement:** All graphical summary plots show individual measures as dots. All primary data are in the supporting tables set up

## Abstract

Interpreting rare variants remains a challenge in personal genomics, especially for disorders with several causal genes and for genes that cause multiple disorders. ZNF423 encodes a transcriptional regulatory protein that intersects several developmental pathways. ZNF423 has been implicated in rare neurodevelopmental disorders, consistent with midline brain defects in Zfp423-mutant mice, but pathogenic potential of most patient variants remains uncertain. We engineered ~50 patient-derived and small deletion variants into the highly-conserved mouse ortholog and examined neuroanatomical measures for 791 littermate pairs. Three substitutions previously asserted pathogenic appeared benign, while a fourth was effectively null. Heterozygous premature termination codon (PTC) variants showed mild haploabnormality, consistent with loss-of-function intolerance inferred from human population data. In-frame deletions of specific zinc fingers showed mild to moderate abnormalities, as did low-expression variants. These results affirm the need for functional validation of rare variants in biological context and demonstrate cost-effective modeling of neuroanatomical abnormalities in mice.

## Author summary

Gene identification in rare disorders is typically supported by finding different mutations of the same gene in multiple families with the same disorder. However, causal evidence for any specific mutation found in one or a few related individuals is weaker, especially if the disorder can be caused by any of several genes and the functional effect of the mutation is not certain. Experimental models can be helpful in testing causal effects, but only

as dataframes should anyone wish to reanalyze them.

**Funding:** This work was supported by grant R01 NS097534 from the National Institute of Neurological Disorders and Stroke to BAH. RZL was supported in part by an institutional grant from the National Institute of General Medical Sciences, R25 GM083275. The funders had no role in study design, data collection and analysis, decision to publish, or preparation of the manuscript.

**Competing interests:** The authors have declared that no competing interests exist.

to the extent that the model is validated to recapitulate one or more aspects of the disorder. We used CRISPR/Cas9-based genome engineering to create a wide range of mutations in mouse *Zfp423*, whose human cognate is implicated in neurodevelopmental disorders, especially cerebellar vermis hypoplasia and Joubert syndrome. This large collection of animal models shows that both reduced *Zfp423* expression, including heterozygosity for loss-of-function mutations, and normally-expressed domain deletions, including specific zinc finger domains, produce measureable abnormalities in midline development. Despite this high level of validation, most patient-derived amino acid substitution variants tested did not produce measureable effects. The single exception is a substitution, H1277Y, that destroys a structural element in the last zinc finger domain and results in dramatic loss of steady-state Zfp423 protein level.

## Introduction

Variant effect prediction remains a challenge in medical genomics [1, 2]. Progress from large reference databases such as ExAC [3], gnomAD [4], and UK Biobank [5] allows powerful statistical evidence against pathogenicity, based on allele frequency [6] for rare variants that had appeared unique to patients in smaller samples. Recessive phenotypes, low or context-dependent penetrance, effects on pre-term viability, and demography, however, may create exceptions often enough to be relevant to patients with rare disorders. Population frequencies also provide limited guidance for singleton and de novo variants. Predictive algorithms based on evolutionary constraint, physico-chemical similarity between residues, or average replacement effects in deep mutational scanning data continue to improve, but these constraints are neither necessary nor sufficient for disease association as loss of human-specific traits may present as disease while evolution selects on subtler variation than disease presentation. Attempts to model prediction accuracy can suffer where ground truth is not available and clinical variant databases in current use include assertions often based on limited evidence. The problem can be particularly acute for disorders where a substantial number of genes are mutable to overlapping phenotypes, including ciliopathies such as Joubert syndrome and related disorders (JSRD). For example, *ZNF423* mutations have been reported as pathogenic in JSRD patients [7] and other neurodevelopmental disorders [8], but most patient variants have uncertain significance and even those asserted pathogenic in public databases rely on very limited data. This is true for many rare disorders.

Mice can be a useful model for *ZNF423* function. The ZNF423 orthology group is highly constrained across vertebrates [9]; after accounting for annotation differences in alternative 5' exons, mouse Zfp423 and human ZNF423 share >98% amino acid identity (99% in zinc finger domains). Most human variants will therefore be in sequence contexts that are similarly constrained in mice. Null mutations in mouse *Zfp423* have defects in midline brain development similar to human JSRD, including anterior rotation of the cerebellar hemispheres and hypoplasia or agenesis of the cerebellar vermis with more modest effects on forebrain structures [10–12]. Roof plate defects in Zfp423 mutant mice also impact development of hindbrain choroid plexus [10, 13]. Other work in mice showed notable effects of Zfp423 on olfactory neurogenesis [14], neocortex development [15], adipogenesis [16–18], and wound healing [19]. Gene-trap alleles that reduced expression of an otherwise normal Zfp423 protein showed hypomorphic phenotypes, indicating a graded response to genetic function [10]. In cerebellum, loss of Zfp423 prevents or limits response to SHH by granule precursor cells ex vivo, consistent with a functional abnormality in the cilium [20]. ZNF423 homologs interact with a

diverse set of lineage-determining or signal-dependent transcription factors in alternate and potentially competing complexes [7, 21–26] and mutational effect will likely depend on which if any of these contacts is altered [27]. Cell-based models may afford screening of all potential variants in a protein [28–30], but the limited context of cells ex vivo could miss key features of ZNF423 function as its expression is dynamic across communicating cell types during development, including both germinal zones in the cerebellum. Whether (or to what extent) all ZNF423 interactions and functions are required in any one cell type is not clear. With the ability to multiplex germline editing at high efficiency and specificity, mouse brain development might therefore be the simplest robust assay for impact of ZNF423 variants on human brain development, allowing quantitative assessment of variant effects on brain development at a scale commensurate with ascertainment of rare disease patient alleles while being agnostic to developmental stage or cell types in which specific functions are compromised.

Here we developed simple, quantitative measures with good statistical power to assess structural brain abnormalities in ~50 mouse strains with *Zfp423* mutations created by genome editing. This extensive set of comparisons allowed us to show differences in sensitivity among phenotypes, test pathogenic potential of patient-derived and other variants, and identify previously unreported haploabnormality in null allele heterozygotes. Among patient-derived substitution alleles, H1277Y, at a zinc-coordinating histidine in the last of 30 C2H2 zinc fingers, was effectively null. By contrast, three other patient variants asserted pathogenic or likely pathogenic based on single patients and algorithmic predictions (R89H, P913L, and E1124K) appeared benign. Premature truncation variants, including humanized alleles that encode protein tails to model patient frame-shift variants, were predominantly null with no evidence for dominant negative activity. An early frame-shift variant in exon 3 was an exception, evading nonsense-mediated decay to produce a partial protein at reduced abundance and a partial loss-of-function phenotype. Null allele heterozygotes showed slightly lower weight, smaller cerebellar vermis, and shorter stride length than control littermates, providing functional evidence for loss-of-function intolerance observed in human population data. In-frame deletions had a range of domain-specific effects. Deletions that remove zinc finger 1, zinc fingers that bind BMP-dependent SMAD proteins, or a non-motif region containing two CXXC sites, and deletions that reduced overall protein expression showed measureable effects, while deletion of zinc finger 12 did not.

## Results

### Induction of patient-derived variants and collateral mutations

We developed an editing pipeline using standard CRISPR/Cas9 tools. We prioritized 13 *ZNF423* amino acid substitution variants and two frame-shifting alleles from patients (S1 Table). Four substitution alleles and one frameshift were published [7, 8] and reported in ClinVar [31] as pathogenic or likely pathogenic; a second frameshift allele was reported in MyGene2 [32]. Other substitution alleles were patient variants of uncertain significance communicated by Drs. Joseph Gleeson and Friedhelm Hildebrandt. Targets were selected to include a range of predicted effects in commonly used variant effect algorithms (Table 1) and to include a range of allele frequencies in databases depleted for close relatives and patients with Mendelian disorders [3, 4]. We edited FVB/NJ embryos because this strain background improved both postnatal viability of the *Zfp423^{nur12}* null mutation and heterozygote breeding performance relative to C57BL/6J, without compromising penetrance of severe ataxia among surviving homozygotes [33]. For most variants, we co-injected two or more alternative repair template oligonucleotides (S2 Table) to create silent substitution controls or edit adjacent sites in a single injection series.

**Table 1. Effect predictions for *ZNF423* substitution variants modeled in this study.**  Variant (domain) shows position relative to NP_ 055884.2, single-letter amino acid codes, and position relative to C2H2 zinc fingers (ZF). Column hg38 shows nucleotide position in the hg38 reference assembly with reference/variant nucleotide. For the gnomAD database (v2), allele count and minor allele frequency (MAF) are given. ClinVar assertions were current at time of writing. Categorical calls and scores are shown for 8 variant effect predictors.

| Variant (domain) | hg38 | gnomAD (MAF) | ClinVar 2019 | PolyPhen2 (HVAR) | SIFT (Score) | PROVEAN | Mutation Taster (rankscore) | Mutation Assessor (rankscore) | VEST3 Rank score | CADD PHRED | Envision |
|---|---|---|---|---|---|---|---|---|---|---|---|
| R89H (ZF1) | 16:49730782 C/T | 18 (6.4 e-5) | Likely pathogenic | Benign (0.001) | Tolerated (0.282) | Neutral (0.19) | Polymorphism (0.261) | Neutral (0.016) | 0.126 | 19.16 | 0.99 |
| G132V (ZF1-2 linker) | 16:49638757 C/A | 0 | - | Damaging (1) | Damaging (0.002) | Deleterious (-2.67) | Disease causing (0.81) | Low (0.225) | 0.883 | 25.6 | 0.85 |
| S382P (ZF8-9 linker) | 16:49638008 A/G | 215 (7.6 e-4) | Uncertain significance | Possibly damaging (0.72) | Tolerated (0.254) | Neutral (-1.65) | Disease causing (0.345) | Low (0.498) | 0.682 | 24.4 | 0.92 |
| R760C (ZF18) | 16:49636874 G/A | 11 (3.8 e-5) | - | Probably damaging (0.82) | Damaging (0.001) | Deleterious (-4.83) | Disease causing (0.588) | Low (0.304) | 0.904 | 28.9 | 0.74 |
| P913L (ZF21) | 16:49636414 G/A | 47 (1.7e-4) | Pathogenic | Probably damaging (0.92) | Damaging (0.010) | Deleterious (-3.87) | Disease causing (0.81) | Medium (0.833) | 0.842 | 28.5 | 0.89 |
| Q1008H (ZF24) | 16:49636128 C/G | 7 (2.8e-5) | - | Probably damaging (0.88) | Damaging (0.000) | Deleterious (-3.61) | Disease causing (0.457) | Medium (0.53) | 0.848 | 25.3 | 0.93 |
| Y1064C (C4) | 16:49635961 T/C | 1 (4.2e-6) | - | Probably damaging (0.75) | Damaging (0.031) | Deleterious (-5.02) | Disease causing (0.548) | Low (0.246) | 0.815 | 28.1 | 0.74 |
| K1071Q (C4) | 16:49635941 T/G | 0 | - | Probably damaging (0.65) | Damaging (0.036) | Neutral (-1.52) | Disease causing (0.465) | Low (0.246) | 0.577 | 28.5 | 0.92 |
| E1124K (ZF26) | 16:49635782 C/T | 8 (3.2e-5) | Likely pathogenic | Possibly damaging (0.54) | Tolerated (0.092) | Neutral (-1.64) | Disease causing (0.548) | Medium (0.562) | 0.883 | 23.6 | 0.98 |
| H1277Y (ZF30) | 16:49491301 G/A | 0 | Pathogenic | Benign (0.04) | Damaging (0.002) | Deleterious (-3.93) | Disease causing (0.53) | Medium (0.924) | 0.951 | 24.5 | 0.92 |

We recovered 12 out of 15 designed mutations–10 of 13 intended substitution alleles and each of two patient frame-shift variants with humanized codons–as well as designed silent control edits and a large number of collateral mutations (Fig 1A, S3 Table) and verified transmission by genotyping (S4 Table). Substitution alleles included sites that are highly conserved among vertebrates, including seven that were invariant among 165 curated orthologs [9]. Most lie in or adjacent to zinc finger (ZF) domains (Fig 1B) or a non-motif region containing a quartet of conserved cysteine residues with potential to form a C4-class zinc finger and for which common annotation tools showed wide disagreement on functional predictions (Table 1). Collateral variants included predicted loss of function (pLOF) alleles at several positions in the coding sequence and in-frame deletions that allowed observations of protein stability and domain-specific function in vivo. We included several variants that remove specific zinc fingers or conserved regions between zinc fingers in subsequent analyses. A simple measure of cerebellar vermis width from surface views of the brain was sensitive to heterozygosity for presumed null alleles and specific to variants that removed critical residues or decreased protein abundance, while comparison among non-mutant littermate pairs suggested high sensitivity and power to detect modest differences in 10–15 sample pairs (Fig 1C, S5 Table, and results below). Importantly, none of the variant effect predictions (Table 1) correctly predicted outcomes with respect to disease-relevant phenotypes (Fig 1C and below).

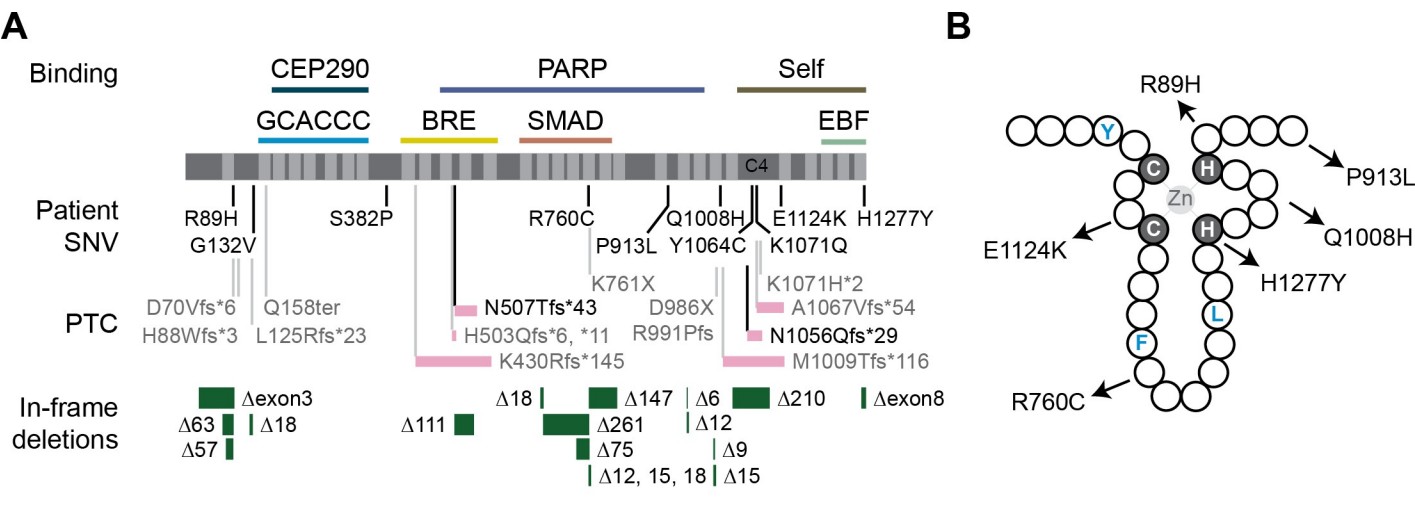

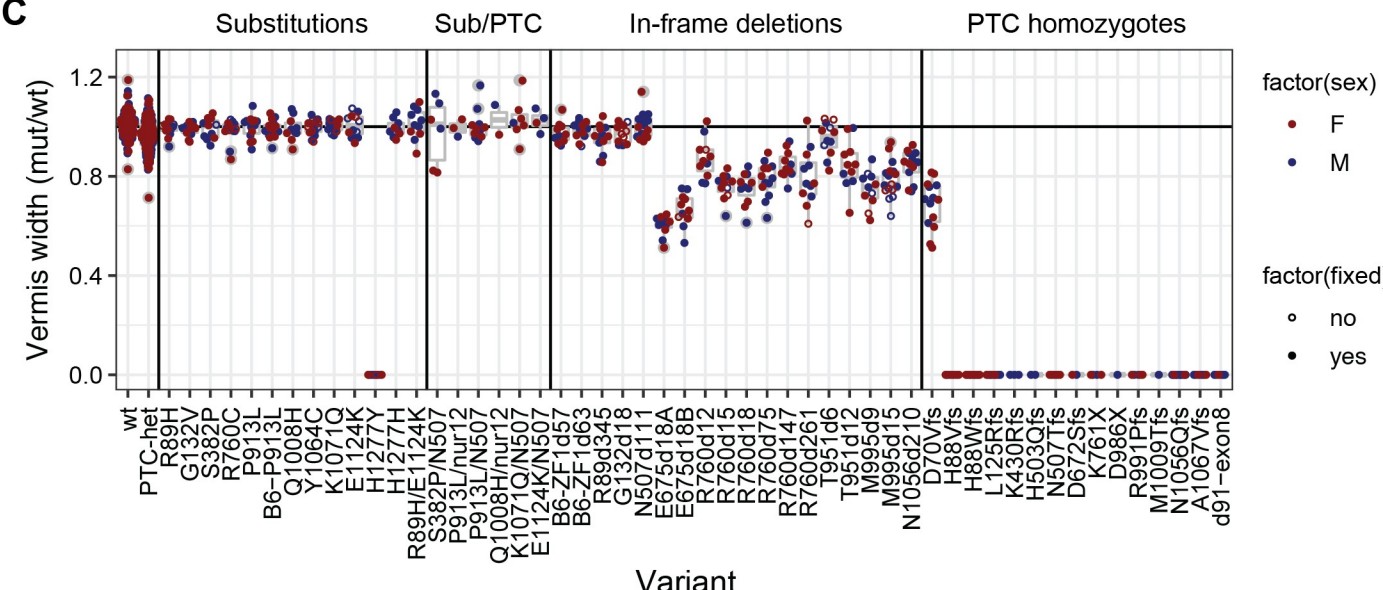

**Fig 1. *Zfp423* mutations induced by genome editing affect vermis size.** (A) Locations of specific variants are indicated relative to ZNF423 RefSeq protein NP_055884 (dark grey), C2H2 zinc fingers (light grey), and reported binding activities. Frame-shift PTC variants show relative length of altered reading frame (pink bars). N507Tfs*43 and N1056Qfs*29 were humanized to encode the same frame-shifted peptide as reported patient variants. For in-frame deletions (green bars), the number of deleted base pairs is shown. (B) Schematic of a C2H2 zinc finger shows relative positions of patient substitution alleles relative to consensus hydrophobic (blue) residues and required zinc-coordinating residues (grey background). (C) Ratio of mutant to control vermis width measured from surface views of same-sex littermate pairs. Each dot represents one littermate pair. Female pairs, red, male pairs, blue. A few pairs were measured as freshly dissected, unfixed material (open circles), which did not noticeably affect the relative measure. PTC heterozygous, H1277Y homozygotes, homozygotes for each in-frame deletion except N507d111, and all PTC homozygous variants were significantly different from both the expected null model and empirical wild-type:wild-type comparisons.

## Simple measures are highly sensitive to *Zfp423* variants

To test variants for pathogenic potential at moderate to large scale, we looked for phenotypes that were easily obtained and robust across trials. We observed home cage behavior and video-graphed mice walking across an open stage to assess gross locomotor activity for each variant subjectively (S1–S4 Videos). We quantified viability, weight, and anatomical parameters of brains in both surface view and block face photographs as quantitative phenotypes (S6 Table). A subset of null-allele heterozygotes was tested in more detail for locomotor behaviors (S7

Table). To avoid potential confounding effects of age, sex, maternal care, or other factors that might vary across a large colony, all tests were conducted on co-housed, same-sex littermates and scored as mutant/control ratio (anatomy) or differences (behavioral latencies). Variants were studied concurrently by investigators blinded to genotypes.

Each measure showed large effects of *Zfp423* premature termination codon (PTC) alleles in homozygotes, with gross ataxia and cerebellar vermis hypoplasia showing complete penetrance. Predicted null variants had no detectable Zfp423 protein on Western blots, with a detection limit ≤5% of control littermate levels, except D70Vfs*6, which showed a reduced level of a lower molecular weight protein unique to that allele (Fig 2A and S1 Fig). For protein-null alleles (PTC variants except D70Vfs*6), homozygotes were recovered at reduced frequency and several null animals identified at P10-P15 died prior to assessment, showing a relatively broad window for lethal events (Fig 2B). Among survivors, locomotor disability was severe and never overlapped controls (S1 Video). None of 72 *Zfp423*-null brains had measureable vermis in surface photographs (Fig 1C). Nearly half of those with mid-sagittal block face views (30/66) showed a small amount of cerebellar tissue, largely due to compression of lateral hemispheres toward the midline (Fig 2C and 2D). Midline fiber tract (corpus callosum and anterior commissure) and cortical thickness measures distinguished mutant from control groups, but with smaller relative magnitude and incomplete penetrance (Fig 2E–2J). Among forebrain phenotypes, corpus callosum had the largest effect and highest penetrance, while anterior commissure effects were only evident across a larger population of animals. PTC homozygotes were smaller than littermates, with lower weight at sacrifice (Fig 2K and 2L). These results generalized findings from earlier studies and put quantitative parameters on magnitude and penetrance for hindbrain, forebrain, and weight measures.

## Loss of function heterozygotes showed reduced vermis, weight, and stride without evidence for dominant negative activity

Having a large set of protein-negative PTC alleles across the full of the coding sequence allowed us to compare presumptive null alleles and distinguish potential haploabnormalities from dominant negative effects. PTC variants whose RNAs escape nonsense-mediated decay often enough to produce detectable levels of variant protein can have dominant negative genetic properties, by decoupling functional domains relative to the non-mutant protein [34]. Early PTCs can result in protein translation from an alternative initiation codon [35, 36] or exon skipping, while late PTCs could produce truncated proteins if they escape nonsense-mediated decay (NMD). *Zfp423* has a highly unusual gene structure, including 4-kb internal coding exon (Fig 1) and variant effects in situ might differ from assays performed in cell culture on compact gene structures according to the proposed "long exon rule" [36, 37]. In Chaki et al. [7], one of us (B.A.H.) speculated that JBTS19 patients carrying one PTC variant heterozygous to an apparently normal allele might have dominant negative activity, supported by transfection assays with a corresponding cDNA in a human cell line. To test this in vivo and to learn more about how PTC variants behave in the context of a very large exon structure, we examined 16 mouse lines carrying different PTC variants at distinct positions relative to exon boundaries and encoded protein domains.

Heterozygotes for PTC variants that do not produce detectable protein should also be a good test of sensitivity, since multiple labs previously reported only recessive phenotypes from several alleles [10–12, 27]. Physical measures were slightly decreased on average in heterozygotes compared to sex-matched littermates across PTC variants for which homozygotes were protein-negative. Vermis width (Fig 3A) and weight at sacrifice (Fig 3B) each had a ~3–4% decrease with strong statistical support ($p<10^{-6}$), while anterior commissure measure had a

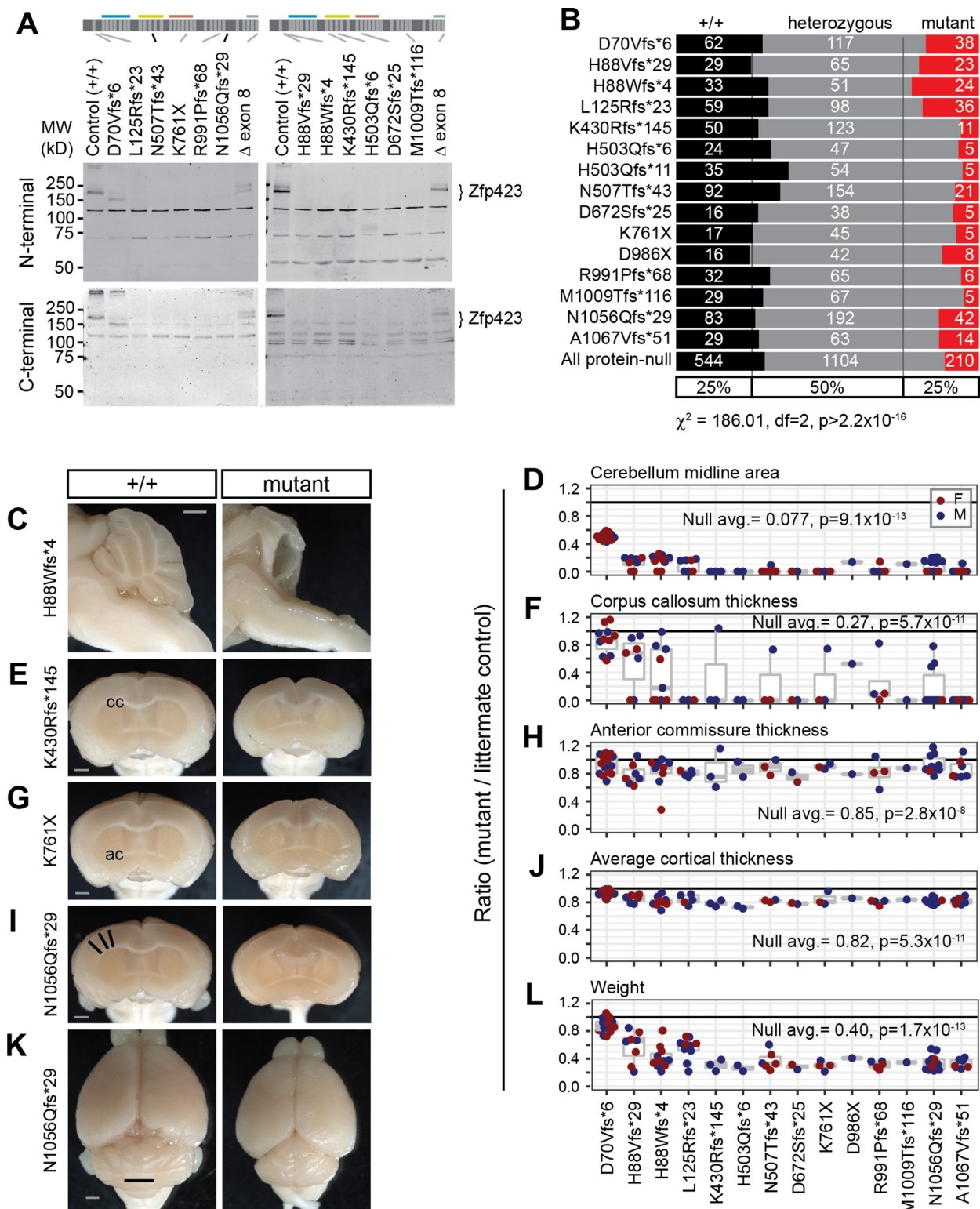

**Fig 2. *Zfp423* frameshift and nonsense mutations are effectively null, except D70Vfs\*6.** (A) Western blots detected full length Zfp423 in perinatal cerebellum of control samples. Among PTC homozygotes, D70Vfs\*6 and a deletion of terminal exon 8 showed altered-size proteins at reduced levels. No consistent evidence for residual protein was seen for any other PTC variants, detection threshold ≤5% wild-type level. N-terminal, Bethyl A304-017A. C-terminal, Millipore ABN410. Cross-reacting background bands were independent of PTC position. (B) Reduced frequency of homozygotes for each PTC at biopsy (P10-P20) from breeding records. Summary chi-square is for protein-null alleles (excluding

D70Vfs*6). (C) Mid-sagittal images showed variable amount of midline cerebellar tissue in mutants, with residual tissue attributable to hemispheres. (D) Ratio of midline cerebellum area (mutant/control) from block face images. Coronal block face images showed abnormal forebrains, including (E, F) disrupted or reduced corpus callosum (cc), (G, H) reduced anterior commissure (ac), and (I, J) reduced cortical thickness, measured as the average radial distance at 15˚, 30˚ and 45˚ from midline (black lines). (K) Representative surface views. Vermis width as plotted in Fig 1 is indicated (black line). (L) PTC mutants had reduced body weight at sacrifice. (D, F, H, J, L) Averages and Wilcoxon signed-rank test p-values for ratio = 1 for combined data from all PTC alleles excluding D70Vfs*6 are shown. Female pairs red, male blue. Scale bars, 1 mm.

1.4% decrease with nominal support (p = 0.046 after Benjamini-Hochberg correction for false discovery). In contrast, average thickness of cortex (p = 0.11) and corpus callosum (p = 0.38)

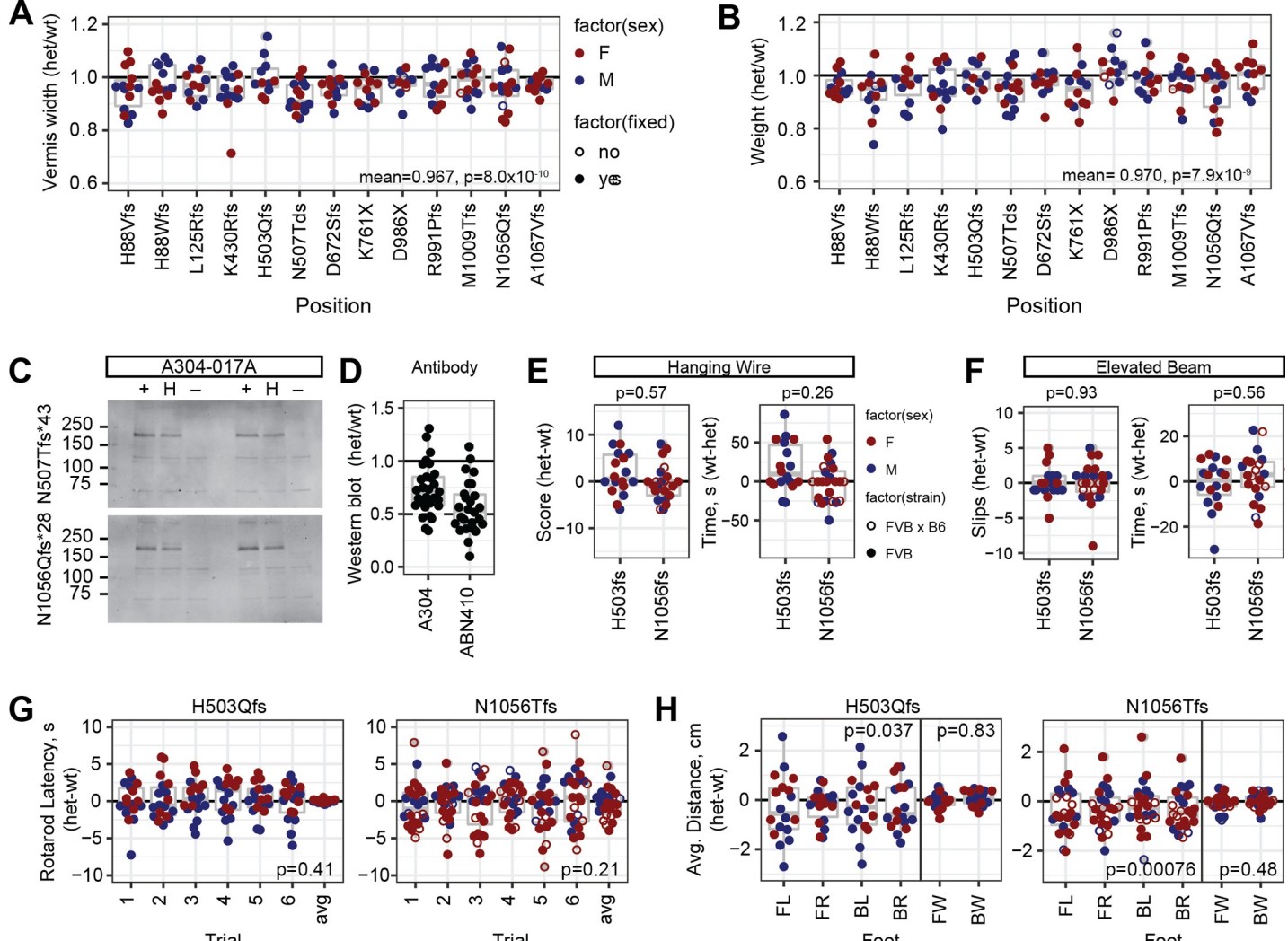

**Fig 3. PTC heterozygotes have mild haploabnormalities.** (A) Vermis width in heterozygotes for individual protein-negative PTC variants relative to control littermates. Y-axis was shifted relative to Figs 1 and 2 to emphasize distribution within the range of observed values. Bottom right corner, mean and p-value for all PTC heterozygotes, combined N = 165, one-sample t-test for true ratio = 1. (B) Weight at sacrifice, combined N = 163. (C) Western blots showed reduced steady-state level of Zfp423 protein in neonatal cerebellum of PTC heterozygotes relative to reference littermate. (D) Normalized values among PTC heterozygotes with independent antibodies showed typical values between 0.5 and 1 relative to control littermates in neonatal cerebellum. (E-H) Behavioral tests for locomotor coordination on littermate pairs for two PTC heterozygotes, H503Qfs*11 (N = 18) and N1056Qfs*29 (N = 24), either coisogenic on FVB (filled circles) or as B6xFVB F1 hybrids (open circles). (E) Hanging wire task performance scores and fall latencies. Wilcoxon signed rank test, N = 43. (F) Elevated beam escape task. (G) Rotating rod fall latencies across six successive trials per pair and average ratio across all six trials. P-value is one-sample t-test for average difference being 0. (H) Footprint analysis for stride length in each paw and width between left and right paws. FL front left, FR front right, BL back left, BR back right, FW front paw stride width, BW back paw stride width. P-values for one-sample t-tests for average differences equal to 0.

showed little evidence for heterozygote effects, perhaps due to the modest effect size on cortex even in null animals and the comparatively high variance on corpus callosum measures.

Infrared fluorescence Western Blots from neonatal cerebellum (Fig 3C and S2A Fig) supported average expression ratios between 0.56–0.73 with two different antibodies (N = 29 comparisons for A304, 26 for ABN410), albeit with substantial variation across experiments (Fig 3D). Expression ratio greater than 0.5 is consistent with a proposed negative autoregulatory activity of Zfp423 [38]; it may also reflect changes in tissue composition of cell types and states, although the strongest source perinatally should be granule cell precursors, which decrease in mutant animals [10, 20]. With these caveats, identification of mild phenotypes in null heterozygotes places limits on the degree of protein functional deficit required to produce phenotypes relevant to disease modeling for structural abnormalities.

Quantified measures of locomotor function, performed by a core facility blind to genotype and experimental goals, supported a modest difference only in stride length for PTC heterozygotes. Joint analysis of two heterozygous variants, H503Tfs*11 (N = 18 littermate pairs) and N1056Qfs*29 (N = 24 littermate pairs), did not identify significant differences by genotype on a hanging wire task for grip strength and coordination (Fig 3E), elevated beam escape (Fig 3F), nor accelerating rotating rod (Fig 3G) tasks for locomotor coordination. Footprint analysis for gait parameters showed ~5% reduced stride length that was independently significant for each variant, although potentially confounded by animal size as assessed by weight, with no detected difference in front or back stride width (Fig 3H). These data supported better sensitivity and cost effectiveness of anatomical measures than simple behavioral measures to perturbations in *Zfp423* function in mice.

## Differential effects near either end of the open reading frame

Despite being created on the more sensitive B6 background, D70Vfs*6 showed milder phenotypes than all other PTC variants, while a 91-bp deletion that removed just the nine terminal residues encoded by exon 8 on the FVB background appeared similar to null variants. Western blots showed that D70Vfs*6 produced a detectable pool of lower-molecular weight Zfp423 protein (Fig 4A and S2B Fig), presumably by translational initiation after the introduced stop codon at position 72 in its first open reading frame. The next available in-frame AUG is at position 118, between ZF1 and ZF2, and would include each of the previously annotated binding domains. The exon 8 deletion produced a protein of near-normal size, but one that must lack both histidine residues from ZF30, which is required for binding EBF proteins [26, 39] and at substantially reduced steady-state level (Fig 4B and S2C Fig). D70Vfs*6 homozygotes have mild to moderate ataxia and hypomorphic anatomical features, most notably vermis hypoplasia (Fig 4C and 4E). In contrast loss of exon 8 had severe anatomical abnormalities, including nearly complete loss of vermis and corpus callosum in all six animals assessed (Fig 4D and 4E). Each of these mutations had significantly reduced frequency of homozygote offspring from heterozygote crosses (Fig 4F). These two mutations showed somewhat unexpected features of *Zfp423* functional organization and reinforced the need for empirical testing of predicted variant effects.

## H1277Y is pathogenic in mice, three other asserted mutations and six VUS are not

We assessed 10 patient-derived substitution alleles: four asserted pathogenic in ClinVar and six rare variants of uncertain significance spanning a range of allele frequencies in public databases from zero to 7.6 x $10^{-4}$ (Table 1 and Fig 1). Each position is highly conserved across vertebrates, except R89H. Eight lie in or adjacent to C2H2 zinc fingers (Fig 1B) or a putative C4

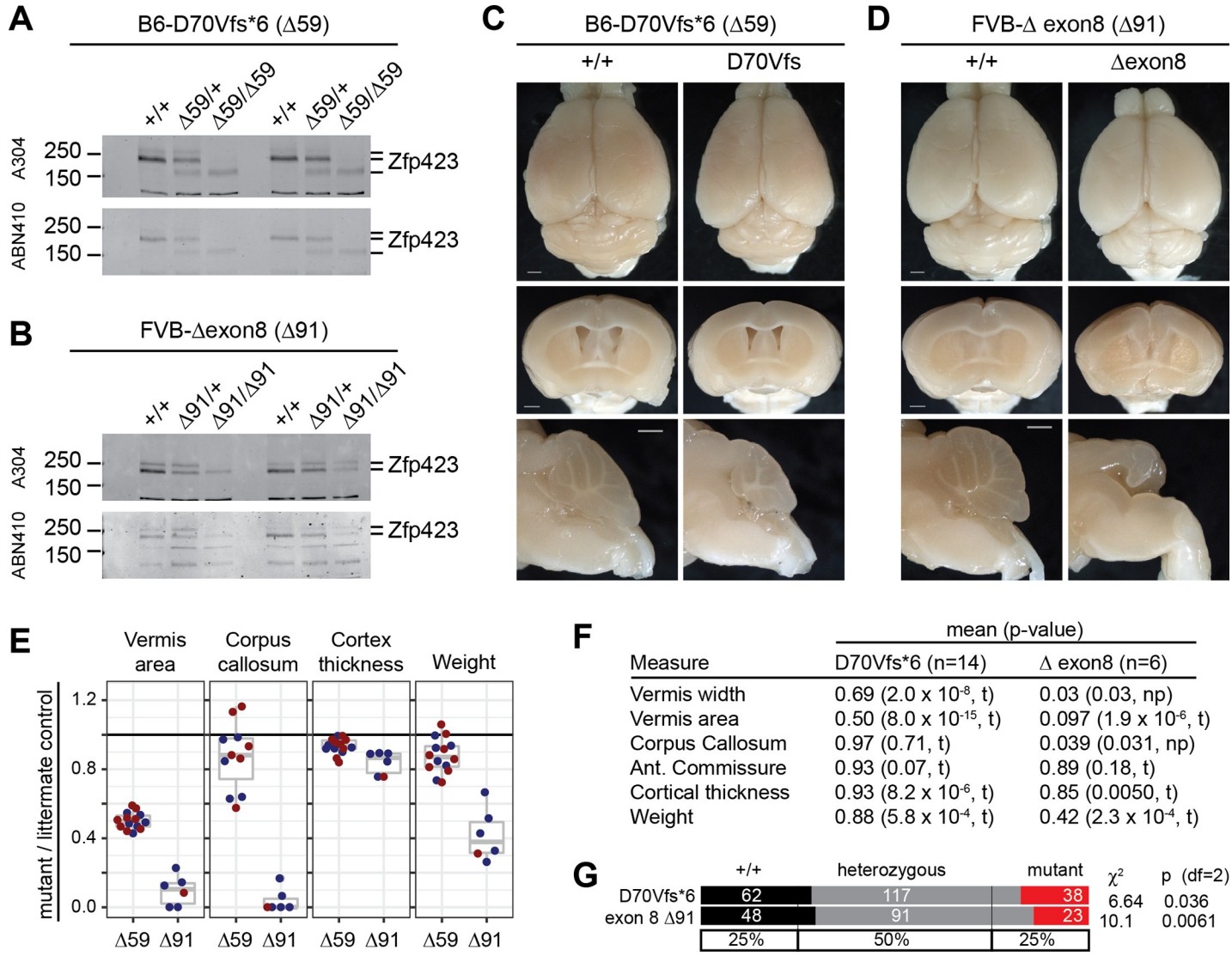

**Fig 4. D70Vfs*6 is hypomorphic while deleting exon 8 is approximately null.** (A) Western blots from neonatal cerebellum from independent trios with two different antibodies against Zfp423 showed a reduced amount of lower molecular weight protein derived from the D70Vfs*6 mutation. (B) Similar blots from two trios for deletion of exon 8, encoding the final nine amino acids of ZF30, showed reduced level of near-full length protein. (C) D70Vfs*6 anatomical phenotypes included reduced cerebellum size. Scale bars, 1 mm. (D) Exon 8 deletion phenotypes approximate those of null alleles despite persistent protein. Scale bars, 1 mm. (E) Quantification of anatomical measures from D70Vfs*6 (Δ59) and exon 8 deletion (Δ91) homozygote and control littermate pairs. (F) P-values from one-sample t-test (t) or non-parametric Wilcoxon Signed Rank test (np) for anatomical measures. (G) Both D70Vfs and exon 8 deletion showed reduced frequencies of homozygotes in crosses.

zinc finger [9]. Among these ten, only H1277Y showed a severe disease-related abnormality while none of the others was distinguishable from control littermates in our assays.

H1277Y was identified in a patient with cerebellar vermis hypoplasia, nephronophthisis, and perinatal breathing abnormalities [7]. By replacing a zinc-coordinating histidine with tyrosine, H1277Y should disrupt the structure of ZF30, which is required for interaction with EBF family proteins [26, 39]. Mice homozygous for the H1277Y variant showed many features of null alleles, including gross ataxia (S2 Video and S3 Video), vermis agenesis, reduced cortical thickness, and incomplete corpus callosum (Fig 5A) while a silent control edit was indistinguishable from control littermates (Fig 5B). Zfp423 H1277Y protein had substantially reduced

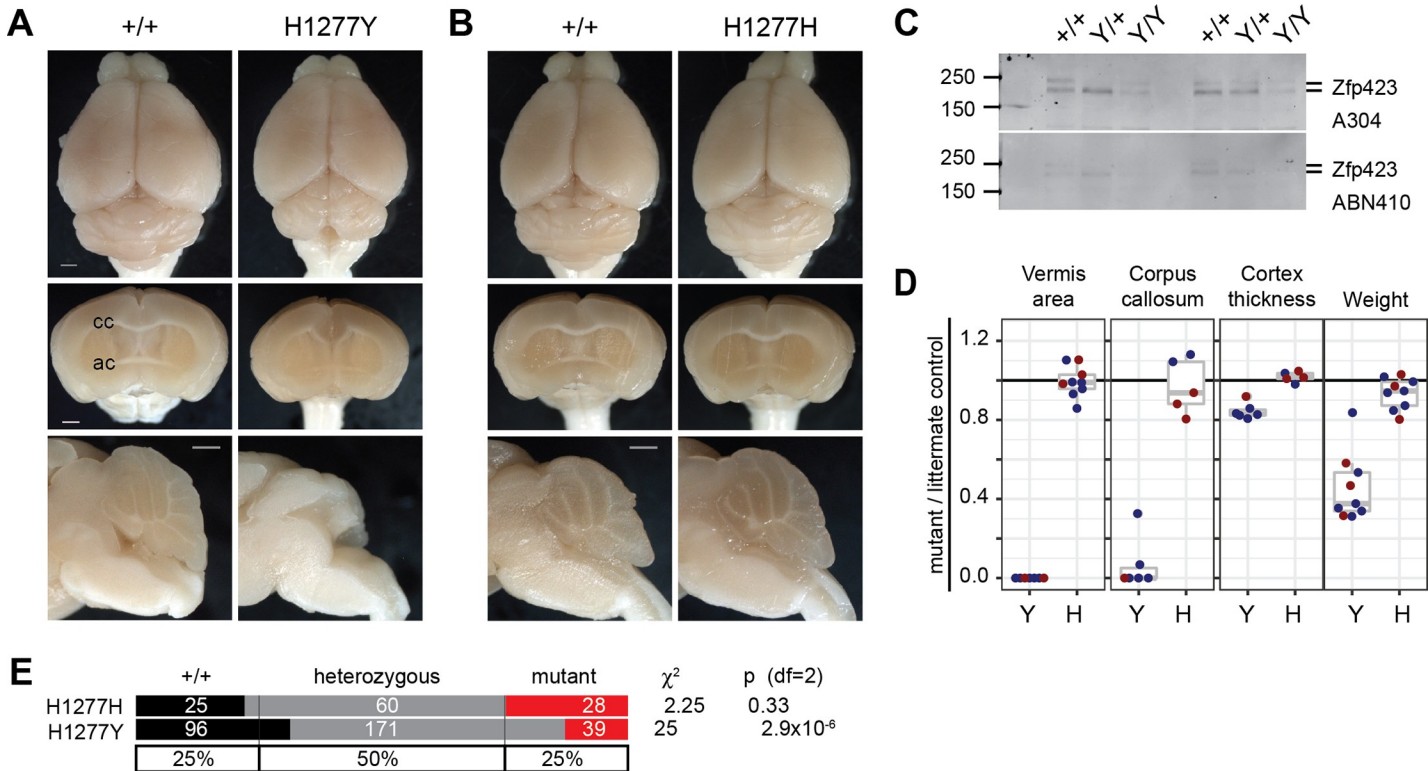

**Fig 5. H1277Y is pathogenic and has reduced protein abundance.** (A) Surface and block face views of same-sex littermates show H1277Y midline defects similar to null alleles, including complete loss of vermis, loss of corpus callosum crossing, and reduced cortical thickness. Scale bars, 1 mm. (B) Silent substitution H1277H is indistinguishable from control littermates. (C) Western blots with antibodies to amino-terminal (top) or carboxy-terminal (bottom) domains show dramatic loss of Zfp423 protein abundance. Results from two distinct trios shown. (D) Ratios relative to control same-sex littermates quantify loss of vermis, midline corpus callosum, cortex thickness, and body weight for H1277Y, but not H1277H. (E) Breeding records show reduced frequency of H1277Y, but not H1277H, homozygotes from heterozygote crosses.

abundance in neonatal cerebellum, similar to the exon 8 deletion and consistent with structural destabilization of the terminal zinc finger (Fig 5C and S2D Fig). Quantitative measures from multiple same-sex littermate pairs show full penetrance of severe defects in surviving mutants of both sexes for cerebellar vermis hypoplasia, loss of corpus callosum at the midline, cortical thickness, and body weight (Fig 5D). Retrospective analysis of breeding records showed reduced frequency of homozygotes for H1277Y, but not the silent substitution control allele (Fig 5E). These results confirm the pathogenic nature of H1277Y for structural brain abnormalities.

The other three asserted pathogenic variants were not sufficient to induce JSRD-like or other obvious phenotypes, nor were any of the tested VUS alleles. P913L was identified by homozygosity in a consanguineous patient with cerebellar vermis hypoplasia, nephronophthisis, and situs inversus [7]. R89H and E1124K were found together in a patient with macrocephaly, extended subarachnoid spaces, and thin corpus callosum [8]. Each of these variants was absent in contemporaneous control subjects. Each was later found in new and larger public databases, but only at low allele frequencies ($1.8 \times 10^{-4}$ to $3.2 \times 10^{-5}$, Table 1) and only as heterozygotes. Using a minimum 10 replicate sample pairs in mice, we did not identify any defect in vermis, midline forebrain phenotypes, or gait for P913L (S3 Fig, S4 Video), nor for R89H or E1124K homozygous mice, nor for R89H/E1124K compound heterozygous mice (S4 Fig). Substitution variants heterozygous to a null allele did not show any effect in smaller sample

sizes. For P913L, we created the same mutation on the more sensitive B6 background, again with no evidence for an effect on brain structures typically affected in *Zfp423* mutants nor reduction in protein level (S1C Fig, S2E Fig, S3 Fig). Pooling all substitution variants except H1277Y and its silent control H1277H to test for a generalized substitution effect at high power (N = 179 littermate pairs) did not produce statistical support for abnormality in any measure before correction for multiple tests (p>0.15 all tests). These data confirmed the pathogenic nature of H1277Y, but supported a more benign interpretation of all other non-synonymous substitutions tested.

## Deletions that remove SMAD-binding fingers or a potential C4-ZF domain produce intermediate alleles

We examined several in-frame deletions (Fig 1A) for Zfp423 protein abundance and brain phenotypes. Recovered examples included distinct Zfp423 functional domains (Fig 6A), several of which had notable reductions in cerebellar vermis (Fig 6B). Several variants with strong phenotypes also significantly affected protein level (Fig 6C, S1C Fig, and S5 Fig) and we could therefore not distinguish between functional requirement for a domain and destabilization of the protein due to awkward breakpoints, although comparison to null heterozygotes suggested that strong phenotypes associated with >75% of control expression levels probably indicates a sequence-specific function.

Cerebellar vermis was the most sensitive anatomic measure to in-frame deletions as a class (Figs 1C and 6D). Impact of intermediate alleles on corpus callosum measures had high variance that limited power after correction for multiple tests (Fig 6E) and lacked effect magnitude for simple measures of cortical thickness (Fig 6F). As a group, in-frame deletions slightly decreased body weight (Fig 6G).

Mutations that substantially reduced protein level also had the strongest effects on vermis measures. Overlapping deletions in ZF15 (E675Δ18A, E675Δ18B) and ZF24 (M995Δ9, M995Δ15), and a single small deletion in ZF18 (R760Δ18), each of which is predicted to destabilize the C2H2 structure by removing critical residues, all reduced protein expression level and reduced cerebellar vermis midline area by approximately half.

Four mutations removed significant protein-coding sequences without reducing measured protein level: G132Δ18 p.del(Leu125-Glu130), which removes 6 amino acids in the sequence between ZF1 and ZF2; N507Δ111 p.del(Arg500-Ile536), which removes ZF12 in the BRE-binding region; R760Δ261 p.del(E675-K761), which fuses ZF15 to ZF18 while deleting ZF16-ZF17 in the SMAD-binding region (R760Δ147 and Δ75 also delete fingers within the SMAD region); and N1056Δ210 p.del(Thr1032-Gly1102)>Arg, which removes part of ZF25 and all of the C4 ZF-like sequence. Surprisingly, deletion of ZF12 (N507Δ111) in the annotated BRE-binding region had no measurable effect. The small deletion between ZF1 and ZF2 (G132Δ18) had a nominal effect (mean = 0.97, p = 0.035, one-sample t-test, N = 12). However, deletions in the SMAD-binding region that did not disrupt C2H2 structural elements and retained near-normal protein levels (R760Δ261, R760Δ147, and R760Δ75) each showed a strong intermediate phenotype, consistent with independent SMAD-domain deletions reported by Casoni et al. [27]. Deletion of ZF25 and putative C4 ZF (N1056Δ210) showed a similar degree of vermis hypoplasia, providing the first evidence for organic function of these domains. These results showed that *Zfp423* brain structural phenotypes were sensitive to most in-frame deletions, often including reduced protein levels, and that different ZF domains or clusters had different degrees of sensitivity. That in-frame deletions were hypomorphic rather than effectively null reinforced the idea that Zfp423 coordinates activities among its interaction partners rather than being an essential component of one pathway.

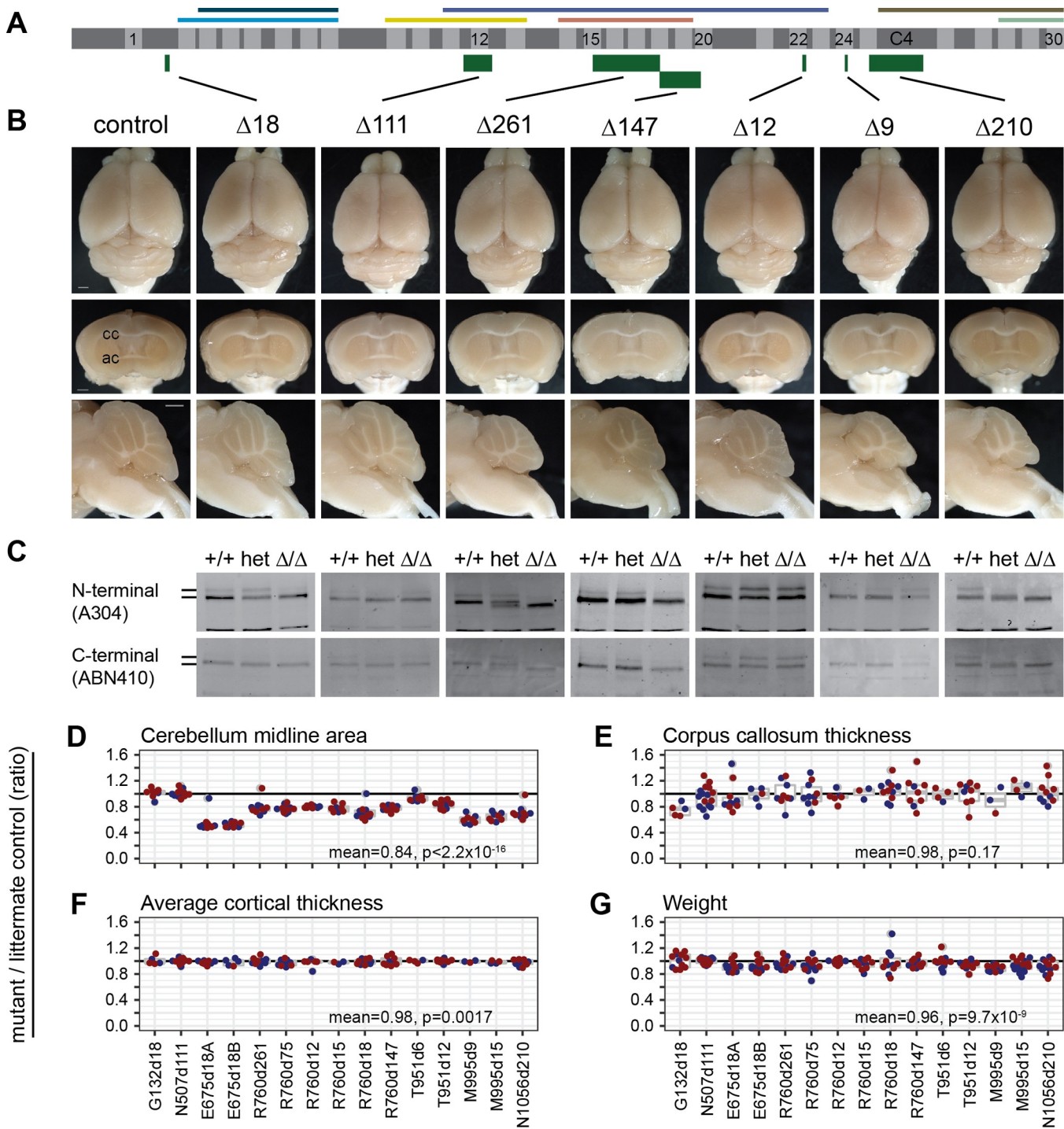

**Fig 6. In-frame deletions that remove critical regions or reduce protein abundance are hypomorphic.** (A) Schematic as in Fig 1 shows the locations relative to 30 C2H2 zinc fingers and known binding regions of in-frame deletions with example data below. (B) Surface views, coronal, and sagittal block face preparations of brains from typical control and in-frame deletions G132Δ18, N507Δ111, R760Δ261, R760Δ147, T951Δ12, M995Δ9, and N1056Δ210. Scale bars, 1 mm. (C) Western blots show Zfp423 proteins in newborn cerebellum for control, heterozygote, and mutant littermates for each variant. (D) Cerebellum midline area, (E) Corpus callosum midline thickness, (F) Cortex thickness as the average of three points at 15°, 30°, and 45° from midline, and (G) Weight at sacrifice each expressed as the ratio of mutant to wild-type same-sex littermate controls for 15 in-frame deletion mutations.

### ZF1 deletion has a reproducibly mild phenotype

The D70Vfs*6 long open reading frame lacks N-terminal residues including the first zinc finger. To determine how much of this phenotype is attributable to loss of ZF1 rather than other N-terminal residues or reduced Zfp423 protein level, we examined three distinct deletions that remove ZF1. One in-frame deletion recovered in the course of modeling patient substitution variants on the FVB background removed all of exon 3 (R89 d345, Fig 1A and 1C), including ZF1 and ~40 other residues. To test the requirement for ZF1 specifically we constructed two smaller deletions (Δ57, Δ63) targeting just the ZF domain in the more sensitive B6 background (Fig 7A). Both targeted alleles expressed Zfp423 protein at levels similar to control littermates (Fig 7B and 7C) and differed only slightly in deletion breakpoints relative to the coding sequence (Fig 7D). All three mutations resulted in slightly smaller cerebellar vermis (Fig 7A and 7E), but less severe than D70Vfs*6 and comparable to reductions seen in null allele heterozygotes. This showed that while ZF1 contributed to Zfp423 function in hindbrain development, it had a smaller effect size than features required for protein production and stability or SMAD binding.

## Discussion

Genomic medicine for rare disorders is often limited by the ability to interpret rare variants. Databases such as ClinVar catalog clinical variants from multiple sources, but report pathogenic assertions based on varied, evolving, and sometimes unclear standards of evidence. Algorithmic predictions and cell models have many benefits, including potential to score all possible single variants, but are prone to errors if the input-output relationships of the assessment do not scale with the impact protein function on the relevant organ system. Attempts to validate effect predictions using clinical variant databases may be somewhat circular if the clinical variants were classified in part on the same criteria as the classifier, such as evolutionary constraint or physico-chemical properties of the substituted residue. Here we showed that current variant effect predictors failed to predict major outcomes for patient *ZNF423* variants accurately. While a minority of variant effect predictors (VEST3, Mutation Assessor) correctly ranked H1277Y as the most likely to be deleterious, neither of these categorically separated this from more benign variants. In contrast, we showed that simple quantitative phenotypes based on domain expertise of a small team can rapidly assess a large array of variants in whole-animal models in a cost-effective manner.

We confirmed H1277Y as a pathogenic variant, while providing evidence against three other variants previously asserted pathogenic or likely pathogenic and six rare/singleton VUSs (Table 1). The quantification of heterozygote phenotypes, as well as inclusion of collateral variants that remove domains or reduce protein abundance, demonstrated sensitivity of simple assays based on prior genetic analyses to relatively modest genetic perturbation. Simple measures with low variance, such as vermis width, allowed adequate sampling for high statistical power and detection of structural changes less than 5% of mean values in the brains in heterozygous animals. This strengthens the interpretation that substitution alleles with no abnormality are benign with respect to major brain phenotypes. By examining in-frame deletion variants, we confirmed the importance of the SMAD-binding ZF cluster, showed that deletion of ZF25 plus a conserved non-motif segment with potential to form a C4-class zinc finger in a region previously implicated in ZNF423 protein self-association is sufficient to cause intermediate decrease in vermis size, and that ZF1 had detectable, but limited, impact on studied phenotypes despite strong conservation across species. Surprisingly, deletion of ZF12 in the BRE-interacting zinc finger cluster had no detectable effect in any of our outcome measures.

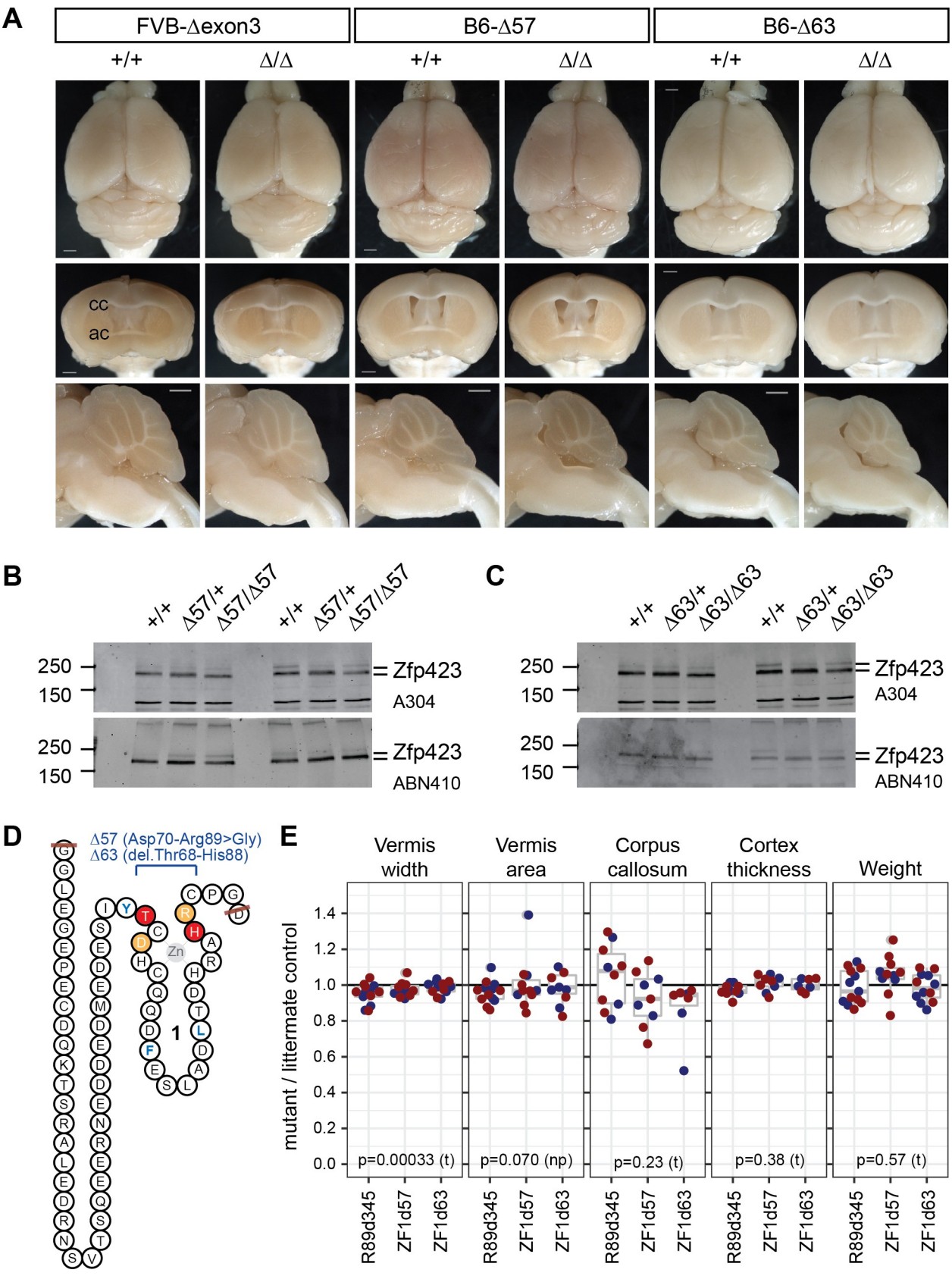

**Fig 7. ZF1 deletions had modest, but measureable impact.** (A) Surface views and block face views from control and mutant same-sex littermate pairs for in-frame deletions removing 345 bp (Δexon3) or smaller fragments within exon 3 (Δ57, Δ63) on FVB or B6 coisogenic backgrounds. Scale bars, 1 mm. (B, C) Western blots show similar levels of Zfp423 protein in control, heterozygote, and mutant newborn cerebellum for Δ57 and Δ63 mutations. (D) Peptide sequence encoded by exon 3, with zinc finger 1 schematized and deletion boundaries indicated in light orange (Δ57) or red (Δ63). Structural hydrophobic residues are blue as in Fig 1B. (E) Ratios between mutant and same-sex littermate controls for vermis width (redrawn from Fig 1C), vermis midline area, corpus callosum midline thickness, average cortical thickness, and weight at sacrifice are shown. P-values are shown for one-sample t-test (t) or non-parametric Wilcoxon signed rank test (np).

## Limitations

While ZNF423-homologous proteins are 98.5% identical in amino acid sequence and previous studies from several laboratories established strong homology between mouse *Zfp423* and human JSRD phenotypes, the degree of sensitivity to mutation could be non-linear. Human and mouse brains, while homologous in structure, develop on different physical and temporal scales. Amino acid substitutions that alter a binding surface could potentially have differential effects depending on conservation of specific binding partners, although this seems unlikely given strong homology of known partners. In order to test many variants rapidly at high power, we focused on simple and less-expensive measures. It is possible that more intensive studies on any specific variant might identify a phenotype, but these are unlikely to be severe in the context of laboratory mice. We also have not examined impacts on other systems, such as olfactory epithelium and adipose tissue, where *Zfp423* null mutants have known phenotypes. While acknowledging these caveats in principle, we nonetheless found high sensitivity of the mouse brain models to even modest genetic perturbation, including heterozygosity for loss-of-function alleles.

## Prospects

Well-powered results from experimental models should inform and modify clinical interpretation of rare alleles. Previous studies identified several variants studied here as pathogenic or likely pathogenic in subjects with JSRD or other neurodevelopmental abnormalities based on being rare variants in a gene with known phenotypic overlap and other properties typically associated with causal variants. However, new evidence should update our expectations and interpretations. For example, P913L was proposed as causal based on its being homozygous in an affected child from a consanguineous pairing, not detected in a control population, and conserved across available vertebrate sequences. Subsequent studies, however, found this allele at modest frequency in the general population, which should reduce confidence in a pathogenic role. In our models, we see no evidence, even on the most sensitive genetic background, for pathogenic consequences of this allele, which should further reduce confidence in a pathogenic interpretation. This variant should now be regarded as likely benign. Similarly, R89H (which is not highly conserved) and E1124K (which is) were reported as likely pathogenic based on being rare or novel in humans and found together in a rare patient. Neither allele when homozygous, nor the two together in trans, nor E1124K in trans to a null allele showed any significant effect on anatomical measures. Interpretation of these variants should also now be modified to likely benign. In contrast, H1277Y, which was strongly predicted to destroy a critical structural residue in the final C2H2 domain, both reduced protein abundance and showed essentially null phenotypes. Interpretation of this variant should now be updated to experimentally supported pathogenic.

More generally, the ability of a single research team to harness domain-specific knowledge to test multiple variants in parallel should improve model-based evidence for causal variants. Our results reinforce the idea that evolutionary constraint is sensitive to much smaller effect sizes than Mendelian disorders and predictions based primarily on constraint are likely prone

to false positive calls. Indeed, seven of ten missense positions studied here were otherwise invariant across 165 diverse vertebrates [9]. Pathogenic variant H1277Y was unique among these in removing a structural requirement for its C2H2 domain, which also reduced protein abundance. A similar logic may limit the predictive value of bulk replacement data from mutational scanning experiments. Incorporating domain-level constraints, like zinc-coordinating residues in zinc fingers, might improve the specificity of useful algorithms. While animal models do not scale sufficiently to test all possible variants, we showed here that mouse models can scale adequately to test plausible variants identified in patients for a rare disorder in order to refine molecular diagnoses. In addition, features that distinguish model-pathogenic from non-pathogenic variants may add to predictive power for untested variants.

## Materials and methods

### Genome editing

All editing experiments used CRISPR/Cas9 ribonucleoproteins (RNPs) based on *S. pyogenes* Cas9. Guide sequences were selected for limited off-target potential using public, web-based tools [40, 41]. Modified (AltR) crRNA guides and tracrRNA were purchased from IDT. Standard and Hi-Fi variant Cas9 proteins were purchased from IDT and New England Biolabs. Single-stranded oligonucleotide donors for homology-dependent repair (Ultramers and Megamers) were purchased from IDT. Injections of FVB/NJ and C57BL/6N one-cell stage embryos were performed in the Rebecca and John Moores UCSD Cancer Center Transgenic Mouse Shared Resource. Guide sequences, predicted scores, RNP assembly conditions, and editing results are given in S1 Table and S2 Table.

### Mutation discovery and validation

Pups derived from injected embryos were screened for developmentally early mutations by PCR-based Sanger sequencing of 500-bp to 700-bp PCR products from crude tail tip lysis DNA preparations. Screening primer sequences are given in S3 Table. Transmission to F1 offspring was confirmed by allele-specific PCR and/or additional DNA sequencing (S4 Table). For mutations with large effects, predicted off-target sites were sequenced to reduce potential for false-positive effects. All variants were also studied across multiple lines and/or backcross generations to further guard against undetected collateral variants by segregation.

### Variant effect prediction

PolyPhen2 [42], SIFT [43], PROVEAN [44], MutationTaster [45], MutationAssessor [46], VEST3 [47], CADD [48], and others were run on VCF files for studied patient variants using wANNOVAR [49] from its web interface (http://wannovar.wglab.org/). A subset of predictions were re-run through their stand-alone web pages for validation. Envision scores [50] were obtained from the Envision web site (https://envision.gs.washington.edu/shiny/envision_new/). Categorical calls (if any) and scores are listed in Table 1. Scales differ across algorithms; original references should be consulted for interpretation.

### Stock maintenance

Mice were maintained by backcross to FVB/NJ or C57BL/6J and by intercrosses to obtain desired genotypes. Mice were maintained in a specific pathogen free (SPF) facility on 12 h light, 12 h dark cycle in high-density racks with HEPA-filtered air and ad libitum access to water and food (LabDiet 5P06).

## Western blots

Cerebellums were manually dissected from young litters (P0-P4) and individually frozen prior to genotyping. Samples from littermate pairs and trios with desired genotypes were homogenized in RIPA buffer supplemented with protease inhibitors (Millipore Sigma P8340) using a small glass dounce (20 strokes). Protein extracts were quantified with BCA assays. 52 µg samples run through Laemmli SDS-PAGE gels before transfer to nitrocellulose membranes (Bio-Rad 1620112). A subset of blots were incubated with Ponceau-S to visualize protein transfer and subjected to image analysis to quantify bulk protein per lane as a reference for subsequent measures. Zfp423 protein was detected using an antibody to residues 250–300 (A304-017A, Bethyl Labs) or residues near the carboxylterminal end of the protein (ABN410, Millipore) with IR-700 conjugated goat anti-rabbit secondary antibody (Rockland 611130122) and detection on a LiCor Odyssey fluorescence imaging station. Processed blots were reprocessed with a cocktail of anti-phosphoprotein antibodies (Millipore Sigma, P3430, P3300, and P3555; with an IR-800 conjugated secondary antibody) as a proxy for total protein. Gel images were quantified in ImageJ and measurements were recorded as background-corrected Zfp423 signal normalized to either Ponceau-S or phosphoproteins.

## Anatomical measures

Samples were prepared, photographed, and measured by an investigator blinded to genotypes. For fixed preparations, deeply anesthetized animals were perfused with phosphate buffered saline followed by 4% paraformaldehyde and brains were removed into fresh 4% paraformaldehyde at 4˚C for 12–24 h and 15–30% sucrose for 24–48 h. Brains were imaged through a dissecting microscope (Zeiss Stemi 2000-C) with a digital camera (Nikon DS-Fi1) using standardized zoom and distance settings and a standard ruler in frame to verify scale. Paired samples were processed together and imaged consecutively. Anatomical features were measured in ImageJ (v1.52a). For surface images, brains were aligned on a swivel-mount platform and photographed dorsal side up. Vermis width was measured at the middle of the folium-tuber lobule (VII). Cerebellar hemisphere height was measured as a vertical line drawn from the dorsal-most point of the simple lobule. Coronal and sagittal block face preparations were made using a standard mouse brain matrix (Zinc Instrument) with the sample aligned anteriorly. Coronal cuts were made at the rostral end of the optic chiasm, through the striatum. Sagittal cuts were made at the midline. Cut brains were mounted on a rotating platform to hold each surface perpendicular to the lens. For cortical thickness, three lines were drawn using the ImageJ ROI Manager, one each at 15 degrees, 30 degrees, and 45 degrees counter-clockwise from vertical, starting at a point where the line would be perpendicular to the angle of the brain surface, and ending at the dorsal side of the corpus callosum; the average of these three measurements was used for each animal. The thickness of the corpus callosum and anterior commissure were measured with vertical lines at the midline and the width of the brain was measured with a horizontal line across the coronal surface at its widest point. Vermis area was measured from midline sagittal block face image using the polygon selections tool in ImageJ to manually define the region of interest.

## Locomotor assays

Gross locomotor function was assessed in home cages and by allowing each test animal and same-sex littermate control to walk freely across a small stage with video recordings for a minimum of three crossings. Rotating rod, footprint pattern, hanging wire, and beam walking tests were performed on same-sex littermate pairs in the Scripps Research Institute Animal Models Core Facility by staff blinded to genotype and hypothesis and following standard protocols.

Animals included roughly equal numbers of male pairs and female pairs for each genotype tested.

**Rotating rod test.**   Latency to fall from an accelerating rotating rod assessed a combination of proprioceptive, vestibular, fine motor, and motor learning capabilities required to avoid falling [51]. Animals were placed on the apparatus (Roto-rod Series 8, IITC Life Sciences, Woodland Hills, CA) prior to acceleration. Latency to fall was recorded by sensing platforms below the rotating rod. Mice were tested in two sets of 3 trials separated by 2 hours.

**Footprint pattern test.**   Footprint pattern analysis assessed basic gait parameters [51–53]. Non-toxic paint was applied to each paw, with front and back paws distinguished by color. Each mouse was placed at one end of a runway covered in paper and allowed to walk until their paws no longer left marks. Forelimb and hindlimb stride lengths (left and right) and front and back leg stride widths were measured the average of three full strides was used for each mouse's values. Mice that did not make 3 measurable strides were excluded.

**Hanging wire test.**   The hanging wire test assessed grip strength and coordination [54, 55]. Mice were held so that only their forelimbs contact an elevated metal bar (2 mm diameter, 45 cm long, 37 cm above the floor) parallel to the ground and released to hang. Each mouse had three trials separated by 30 seconds. Each trial was scored 0 (mouse fell off), 1 (hung onto the wire by two forepaws), 2 (hung onto the wire by two forepaws and attempted to climb onto the wire), 3 (hung onto the wire by two forepaws plus one or both hindpaws), 4 (hung onto the wire by all four paws plus tail wrapped), or 5 (escaped to the ring stand holding the bar or climbed down the stand to the table). Latency to falling off was measured up to a maximum of 30 s.

**Elevated beam test.**   Escape latency and observed foot slips during escape on a narrow beam further assessed locomotor coordination [51]. Three successive trials were recorded. Average escape time and total number of slips were compared between genotypes.

## Statistical analyses

Target samples sizes were estimated from literature and refined according to power calculations based on the observed standard deviations for wild-type littermate pairs as an empirical null model. Retrospective analysis was performed in the R package pwr (v1.2–2, https://github.com/heliosdrm/pwr) or stand-alone software G*power (v3.1.9.2, [56]). Hypothesis tests were performed in R (v3.5.1 [57]). A one-sample t-test was performed for same-sex littermate ratios = 1, or the non-parametric Wilcoxon Signed Rank test if the data distribution showed significant departure from normality by the Shapiro-Wilk test. Multiplicity corrections for family-wise error rate and false discovery rate with mutant classes and false discovery rate across all genotypes are given in S5 Table. Graphical output was in R base graphics or ggplot2 (v3.1.0 [58]) with ggbeeswarm (v0.6.0 https://github.com/eclarke/ggbeeswarm).

To estimate statistical power, we first analyzed non-mutant same-sex littermate pairs, for which data accumulated more quickly than for any specific variant. Vermis width and cortical thickness measures both showed paired sample ratios with low variance and approximately normal distribution. Power calculation for a one-sample t-test estimated 90% power to detect a 10% difference with 5–6 paired samples for a nominal alpha = 0.05 or 10–12 samples after Bonferroni correction for ~50 variants tested for each measure. Other measured values had larger variance (and in some cases incomplete penetrance) and required transformation to meet normality. Vermis width was also the most sensitive measure for variants with any effect (see above). Based on these observations, we set vermis as the primary outcome and 10 sex-matched littermate pairs as a target minimum sample size for testing quantitative effects of *Zfp423* variants.

### Ethics statement

All animal experiments were approved by the University of California San Diego Institutional Animal Care and Use Committee (UCSD-IACUC) under protocol S00291.

### Supporting information

**S1 Table. Embryo injections and recovered mutations.** Cas9 editing formulations and frequency of recovered founders are given.
(XLSX)

**S2 Table. Single-stranded donors used for homology-dependent repair (HDR).** Ultramer and Megamer sequences. Essential variants are shown in red, silent substitutions intended to bias repair or improve genotyping in blue.
(XLSX)

**S3 Table. Screening primers.** Cas9 guide sequences, target sites, and PCR primers used for screening and sequencing potential founders.
(XLSX)

**S4 Table. Genotyping assays and strains deposited to MMRRC.** Primers sequences, restriction enzymes if needed, product sizes, and gel conditions for all variants analyzed. MMRRC stock numbers are included for nine strains accepted by the repository.
(XLSX)

**S5 Table. Nominal and corrected p-values for each genotype.** One-sample tests of mutant/control ratios for each mutation and seven phenotypes. Family-wise error rate (FWER) and false discovery rate (FDR) corrections are listed within each mutational class (i.e., substitutions, in-frame deletions, premature termination codons) for each phenotype. False discovery rate across all genotypes for a phenotype are also listed.
(XLSX)

**S6 Table. Physical measures of sex-matched littermate pairs.** Primary physical measures for each tested animal, organized by littermate pair.
(XLSX)

**S7 Table. Locomotor measures sex-matched littermate pairs.** Primary behavioral measures for each tested animal, organized by littermate pair.
(XLSX)

**S1 Fig. Related to Fig 2. Western blot loading controls and quantification.** (A) Full images for N-terminal blots in Fig 2A. A304-017A antibody detects Zfp423 (black arrowhead) and a variable conformational isomer (gray arrowhead) in control samples as well as a D70Vfs*6-specific protein (purple arrowhead). Molecular weight (kDa) of size marker bands is shown to the left. Membranes re-probed with a cocktail of antibodies againt phosphoserine, phosphothreonine, and phosphotyrosine show approximately even loading. The major band difference in D70Vfs sample was shown in other blots to be a difference between B6 and FVB strain backgrounds. Ponceau-S staining of the membrane before antibody application also shows approximately equal loading. (B) Western blots and stained membranes used for C-terminal antibody ABN410. (C) Approximate quantification by infrared imaging of all blots in this work. All measures adjusted to loading controls and plotted as ratio to wild-type control sample on the same membrane. Dots are measure from A304-017A, crosses from ABN410. Colors indicate genotypes with non-mutant controls in black, heterozygotes in brown and

homozygous mutants in red.
(TIF)

**S2 Fig. Related to Figs 3–5. Western blots and loading controls.** Full blots and Ponceau-S stained membranes for blots shown in Fig 3C (A), Fig 4A (B), Fig 4B (C), 5C (D) and S3C Fig. Size marker molecular weight in kDa is shown to the left. Position of the primary Zfp423 band is indicated by a black arrowhead to the right of the blot and the inconsistent conformational isomer by a gray and any consistently observed mutant specific band is indicated by a purple arrowhead.
(TIF)

**S3 Fig. Related to Fig 5. P913L is not pathogenic in mice.** (A) Surface views of brains from control and mutant same-sex littermate pairs shows grossly normal brains for P913L substitution allele edited independently on FVB/NJ and C57BL/6 (B6) strain backgrounds. (B) Forebrain images show apparently normal structure for P913L mutant on both backgrounds, while highlighting different extent of lateral ventricles between strains at this place of section. (C) Western blots for FVB-P913L (top) or B6-P913L (bottom) with antibody against residues 250–300 (A304-017A) show normal Zfp423 protein abundance. Results from two distinct trios shown. Ratios between same-sex littermates for (D) vermis area at midline, corpus callosum thickness at midline, average cortical thickness at 15˚, 30˚ and 45˚ from midline, and body weight fail to identify defects in P913L homozygotes. Scale bars, 1 mm.
(TIF)

**S4 Fig. Related to Fig 5. R89H and E1124K are not pathogenic in mice.** (A) Dorsal surface, coronal forebrain, and sagittal hindbrain views from control and mutant same-sex littermate pairs showed grossly normal brains for R89H homozygous, E1124K homozygous, or R89H/E1124K compound (trans) heterozygous animals on FVB/NJ background. Ratios between same-sex littermates for (B) vermis midline area, (C) average cortical thickness, (D) midline corpus callosum thickness, and (E) weight at sacrifice fail to identify significant deviations for any of these genotypes. Scale bars, 1 mm.
(TIF)

**S5 Fig. Related to Fig 6. Western blots and loading controls.** Full blots and loading control images for in-frame deletion variants G132Δ18 (A), N507Δ111 (B), R760Δ147 and R760Δ261 (C), M995Δ9 (D), T951Δ12 (E), and N1056Δ210 (F). Size marker molecular weight in kDa is shown to the left. Position of the primary Zfp423 band is indicated by a black arrowhead to the right of the blot and the inconsistent conformational isomer by a gray and any consistently observed mutant specific band is indicated by a purple arrowhead.
(TIF)

**S6 Fig. Related to Fig 7. Western blots and loading controls.** Full blots and loading control images for 57-bp (A) and 63-bp (B) in-frame deletions of zinc finger 1. Size marker molecular weight in kDa is shown to the left. Position of the primary Zfp423 band is indicated by a black arrowhead to the right of the blot and the inconsistent conformational isomer by a gray and any consistently observed mutant specific band is indicated by a purple arrowhead.
(TIF)

**S1 Video. Ataxic gait in N1056Qfs*29 homozygote compared with sex-matched littermate control.**
(MP4)

**S2 Video. Ataxic gait in H1277Y homozygote compared with sex-matched littermate control.**
(MP4)

**S3 Video. Normal gait in H1277H control mutant compared with sex-matched littermate control.**
(MP4)

**S4 Video. Normal gait in P913L homozygote compared with sex-matched littermate control.**
(MP4)

## Acknowledgments

We thank Professors Joseph Gleeson and Friedhelm Hildebrandt for communicating anonymized patient variants and Dr. Kevin Ross for comments on a draft manuscript and Professor Amit Majithia for helpful conversations. We thank families who shared their variants in MyGene2 and ClinVar. We thank Ella Kothari and Jun Zhao for assistance with mouse zygote injections. We thank Dr. Amanda Roberts for advice and coordination of behavioral tests. We thank Abraham Ramirez for excellent animal care.

## Author Contributions

**Conceptualization:** Bruce A. Hamilton.

**Data curation:** Ojas Deshpande, Raquel Z. Lara, Oliver R. Zhang, Dorothy Concepcion, Bruce A. Hamilton.

**Formal analysis:** Bruce A. Hamilton.

**Funding acquisition:** Bruce A. Hamilton.

**Investigation:** Ojas Deshpande, Raquel Z. Lara, Oliver R. Zhang, Dorothy Concepcion, Bruce A. Hamilton.

**Methodology:** Ojas Deshpande, Raquel Z. Lara, Oliver R. Zhang, Bruce A. Hamilton.

**Project administration:** Bruce A. Hamilton.

**Supervision:** Bruce A. Hamilton.

**Validation:** Ojas Deshpande.

**Visualization:** Ojas Deshpande, Bruce A. Hamilton.

**Writing – original draft:** Bruce A. Hamilton.

**Writing – review & editing:** Ojas Deshpande, Raquel Z. Lara, Oliver R. Zhang, Dorothy Concepcion, Bruce A. Hamilton.

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
