## [Decision Letter · Decision Letter 0]

10 Jun 2020

Dear Dr Hamilton,

Thank you very much for submitting your Research Article entitled 'ZNF423 patient variants, truncations, and in-frame deletions in mice define an allele-dependent range of midline brain abnormalities' to PLOS Genetics. Your manuscript was fully evaluated at the editorial level and by independent peer reviewers. The reviewers appreciated the attention to an important topic but identified some aspects of the manuscript that we think should be taken under consideration for improvement during the preparation of a revision.

We therefore ask you to modify the manuscript while considering the review recommendations before we can consider your manuscript for acceptance. Your revisions should address the specific points made by each reviewer. We would suggest careful consideration of some of the comments on potential improvements to the clarity of some of the graphical data presenting in this version.

[LINK]

Yours sincerely,

Rolf W Stottmann

Guest Editor

PLOS Genetics

Gregory Barsh

Editor-in-Chief

PLOS Genetics

Reviewer's Responses to Questions

**Comments to the Authors:**

Reviewer #1: The paper submitted by Prof. Hamilton and collaborators describes a comprehensive collection of around 50 mutations in Zfp423, engineered in mice by genome editing, and their phenotypic outcome. Some of the mutations were previously found in patients with neurodevelopmental defects, and some are instead chosen based on their expected effect on the protein function. Mouse has been chosen as an experimental model since ZNF423 and Zfp423 are highly conserved between human and mice (and across all vertebrates). Therefore, this animal model represents a useful, cost-effective tool to validate the effect of gene variants in complex, developing biological systems compared to cell-based models.

The article is interesting both in the specific ZNF423 characterization and, more broadly, in the way it depicts the potential of the engineered animal model tool. Results are overall convincing, and the substantial load of work conducted to gather all the required information is remarkable. However, the language used to throughout the paper is often unclear and suboptimally suited to a scientific context. Major English editing is therefore needed in order to ensure clarity.

Minors:

Abstract: “In-frame deletions of select zinc fingers…” should be “In-frame deletions of selected zinc fingers…”

Page 3: “Progress from large reference databases such as ExAC [3], gnomAD [4], and UK Biobank [5] allow powerful statistical evidence against pathogenicity…” should be “Progress from large reference databases such as ExAC [3], gnomAD [4], and UK Biobank [5] allows powerful statistical evidence against pathogenicity…”

Page 6: “…allowed additional probes of protein stability…” should be rephrased.

Page 7: “…to detect modest differences in from 10-15 sample pairs…” should be “…to detect modest differences in 10-15 sample pairs…”

Page 9: “To avoid potential confounding of age…” should be “To avoid potential confounding effects of age…”

Page 10: “…in surface photographs (Figure 1D).” Figure 1D doesn’t exist; does it refers to figure 1C?

Page 12: “PTC variants that escape nonsensemediated decay often enough to produce a variant protein can have dominant negative properties by decoupling functional domains [33].” It is quite unclear and should be rephrased

Page 12: “In Chaki et al. [7], one of us (B.A.H.) speculated that JBTS19 patients…” can be better rephrased.

Page 15: “The exon 8 deletion produced a nearly full length protein, but one that must lack both histidine…” can be better rephrased

Figures:

Fig 1B: blue/black are difficult to distinguish

Fig 1C: the figure caption reports “significantly” but no p-value is shown or reported.

Fig 2A: all the mutations should be carried in homozygous fashion, but it is better to reiterate this concept in the caption

Fig 4B: the WT seems to have the same isoform carried by the homozygous delta91 (and in higher amount compared to the other); any explanation for that?

Fig 7D: pink/red and blue are very difficult to spot

WBs in general: in materials and methods it is explained how quantification and normalization are carried out, but it is critical to have either a quantification graph or a loading control image in each panel (or at least in supplementary figures)

Possible in-depht analysis:

in Fig 3A, mutations H88Wfs and H503Qfs seem to have a sex-dependent phenotype (males show mostly and ratio >1, while females show mostly a ratio <1). The same applies to Fig. 3B, for mutations L125Rfs and A1067Vfs (females show mostly a ratio >1, while for males it is always < or around 1). Do the authors have any further evidence of this?

in Fig 3D, animals with a normalized dosage of around 0,5 showed a worse phenotype compared to those with a normalized dosage of around 1?

Did the authors performed any co-IP to prove the retained or lost interaction with known interactors at each mutated site? (e.g., a microdeletion in the SMAD binding domain could or could not affect the SMAD binding and therefore the presence or lack of phenotype, respectively)

Reviewer #2: In this report, Deshpande et al generate a substantial number of mouse strains with mutations in the Znf423 locus. These mutations include known or likely pathogenic variants identified in genomic analysis of patients with Joubert Syndrome and related disorders, as well as variants of uncertain significance. Additional mutant strains were analyzed with various premature termination/frameshift mutations and in-frame deletions. The team leverage the capacity of CRISPR-mediated mutagenesis to generate these models.

The overall impetus of this study is multifold. The efforts leverage existing knowledge of the null phenotypes from previous Znf423 knockout studies to identify somewhat superficial but sensitive and rapidly scorable phenotypes (eg size of the vermis) as a means of determining the phenotypic severity of a given mutation. This is a logical means for quickly assessing phenotypes, which is necessary for the scale of the number mutant lines being studied. From this work the team identify multiple patient-derived variants as being deleterious in the animal model, and other variants as non-phenotypic. Alleles that generate domain-specific deletions also highlight potential ZNF423 protein interaction partners (eg SMADs) and the context-dependency on these interctions. The use of western blotting to assess protein stability is another important feature of the report that provides important information on the nature of different variants. The report will logically lead to further studies in the phenotypes of many of the models generated, which will be an important further studies for Joubert Syndrome.

This substantial effort also highlights the capacity for rapid (relatively) generation of these mouse models and the capacity for relevant high level phenotypic analysis to assess the pathogenicity of many clinically identified variants. This is an important advancement in studies of novel and rare genetic variants identified in human genomics efforts for pathogenicity and causation. There has been concern in some areas of the functional genomics field in the use of the mouse for assessment of variant function; this work is an excellent case study in the scalable capacity for using the mouse in this context. This study will be both of interest to those studying ZNF423 and Joubert Syndrome and also for geneticists with interest in advancing functional genomics studies.

Overall the manuscript is complete and well-written, and the conclusions are consistent with the data presented. Some minor clarifications would be beneficial for the broader readership. Some suggested minor edits:

1. Figures in general.

a. The figures are dense and packed with data. Some smaller panels suffer from a lack of clarity due to their small size. Specific issues for each figure are noted below. The authors may want to consider summarizing the quantitative data and presenting only some representative images, rather than presenting a panel for each mutant cerebellum, for example. Some of the more extensive panels can be shifted to Supplemental data.

b. An image or diagram indicating where the vermis measurement is done would be helpful

2. Supplemental data in general. The Supplemental data contains a substantial amount of fairly raw data. Although useful, some of the data is somewhat too raw and contains lab-specific jargon that makes it difficult to understand. For example, in Table S1, the Results column contains a lot of raw notes on the injection outcome that should be clarified and made more consistent. It is not clear what the difference between ‘edited’ and ‘mutated’ is, and this should be clarified. ‘NTBR’ is not defined. These should be cleaned up and made more clear.

3. Fig 1C – Please clarify that most of the graph consists of analyses of heterozygotes

4. Fig 2B – report individual Chi squares for each genotype offspring data for each mutation

5. Fig 2D clarify in the figure legend that these ratios measures are of homozygous mice.

6. Fig. 3A,B. It would be helpful if individual significance calculations (p value) for each mutant, perhaps in a supplemental table.

7. Fig 3. The reported increased in protein level in hets suggests either negative regulatory loop, or alteration of cell type proportions in het mice. Although a minor point, additional supporting experiments would be helpful. For example, RNA from brains could be examined for transcript steady state levels of RNA from wt allele.

8. Fig 4B. the western is not particularly convincing, partly due to the small size of the panel. Additionally, multiple non-specific bands are apparent, and it appears that multiple bands are affected in the homozygous mutants, including a band the authors do not indicate is ZFP423. The presence of a lane which includes lysate from a bonafide null brain, such as a lysate from one of the several frameshift or nonsense mice used in Fig 2A, would allow the reader to more clearly identify which bands are non-specific, and which are indicative of ZFP423.

9. Fig. 6D. Individual statistical measures of significance should be reported. Given that the authors suggest some alleles are more severe and correlate with protein levels, these individual measures should be indicated.

10. Materials and Methods. Clarify how the brain tissues were fixed, when used, for the gross brain measures. The figure legend of Fig 1 indicates both fresh and fixed tissues were analyzed.

Reviewer #3: In this paper, the authors tackle a key point in personal genomics, namely the reliability of in silico predictions regarding a mutation's clinical impact based on criteria including evolutionary constraints or the physic-chemical consequences of aminoacid substitutions.

They apply their validation strategy to the study of ZNF423/Zfp423, a gene of high relevance in cerebellar development that has been implicated in rare cases of Joubert syndrome, cerebellar vermis hypoplasia and extraneural disorders, including nephronophthisis. The gene also plays key roles in adipocyte development and encodes a protein that acts both as a transcription factor and as a hub for the assembly of multiprotein complexes implicated in key developmental pathways. The protein plays important roles in the regulation of ciliary functions and DNA damage repair.

Contrary to the popular trend of analyzing the effects of mutations in human IPSC-derived neurons, often of an incompletely defined nature, isolated from their extracellular matrix environment, cell-cell interactions and circuit connections, and prone to irreproducible behaviors, they take a systematic in-vivo approach creating a large allele series in the mouse by CRISPR-Cas9-based genome editing. To gauge the effects of these mutations, they resort to simple and significant parameters, which include the size of the cerebellar vermis and of forebrain commissural tracts. In addition, they use motor coordination tests to charaterize mouse behavior.

This paper sets an innovative standard for the systematic functional analysis of gene variants. I expect that it will be widely cited and that it may constitute a cornerstone in the field of functional genetics.

The experiments described in this paper are sound, well presented and appropriately controlled. The results are of very high quality. The paper is well written and the results are thoroughly, yet concisely discussed.

I have no major criticism to express. As a minor point, I suggest that the authors spell the mouse protein in all capitals, which is more appropriate for a genetics journal. Fortunately, in this case, human and mouse homologs have different acronyms.

**Have all data underlying the figures and results presented in the manuscript been provided?**

Reviewer #1: None

Reviewer #2: Yes

Reviewer #3: Yes

PLOS authors have the option to publish the peer review history of their article (what does this mean?). If published, this will include your full peer review and any attached files.

Reviewer #1: No

Reviewer #2: No

Reviewer #3: Yes: Gian Giacomo Consalez

---

## [Decision Letter · Decision Letter 1]

29 Jul 2020

Dear Dr Hamilton,

We are pleased to inform you that your manuscript entitled "ZNF423 patient variants, truncations, and in-frame deletions in mice define an allele-dependent range of midline brain abnormalities" has been editorially accepted for publication in PLOS Genetics. Congratulations!

Yours sincerely,

Rolf W Stottmann

Guest Editor

PLOS Genetics

Gregory Barsh

Editor-in-Chief

PLOS Genetics

Comments from the reviewers (if applicable):

Reviewer's Responses to Questions

**Comments to the Authors:**

Reviewer #1: The article has been improved and all of our comments have been properly addressed. Additional information included in the supplementary figures and tables strengthen the results and provide concrete evidence for the claims made in the paper.

The article is therefore suitable for publication.

Reviewer #2: The authors have addressed the bulk of my concerns, to the best of their capacity given the COVID shutdown.

Reviewer #3: The authors have addressed all points raised by the reviewers, further improving the quality of this exciting paper.

**Have all data underlying the figures and results presented in the manuscript been provided?**

Reviewer #1: Yes

Reviewer #2: Yes

Reviewer #3: Yes

PLOS authors have the option to publish the peer review history of their article (what does this mean?). If published, this will include your full peer review and any attached files.

Reviewer #1: No

Reviewer #2: No

Reviewer #3: No

**Data Deposition**

http://datadryad.org/submit?journalID=pgenetics&manu=PGENETICS-D-20-00765R1

**Press Queries**

---

## [Editor Report · Acceptance letter]

8 Sep 2020

PGENETICS-D-20-00765R1 

ZNF423 patient variants, truncations, and in-frame deletions in mice define an allele-dependent range of midline brain abnormalities 

Dear Dr Hamilton, 

We are pleased to inform you that your manuscript entitled "ZNF423 patient variants, truncations, and in-frame deletions in mice define an allele-dependent range of midline brain abnormalities" has been formally accepted for publication in PLOS Genetics! Your manuscript is now with our production department and you will be notified of the publication date in due course.

With kind regards,

Matt Lyles

PLOS Genetics

On behalf of:
